statistics

science policy, replication, forecasting

**Author for correspondence:**
Thomas Pfeiffer
e-mail: t.pfeiffer@massey.ac.nz

†The two first authors contributed equally to this work.

# Are replication rates the same across academic fields? Community forecasts from the DARPA SCORE programme

Michael Gordon[1,†], Domenico Viganola[2,†], Michael Bishop[4], Yiling Chen[5], Anna Dreber[6,7], Brandon Goldfedder[8], Felix Holzmeister[7], Magnus Johannesson[6], Yang Liu[9], Charles Twardy[3,10], Juntao Wang[5] and Thomas Pfeiffer[1]

[1]New Zealand Institute for Advanced Study, Massey University, Auckland, New Zealand
[2]Department of Systems Engineering and Operations Research, and [3]C41 & Cyber Center, George Mason University, Fairfax, VA, USA
[4]Michael Bishop Consulting, Ottawa, Canada
[5]John A. Paulson School of Engineering and Applied Sciences, Harvard University, Cambridge, MA, USA
[6]Department of Economics, Stockholm School of Economics, Stockholm, Sweden
[7]Department of Economics, University of Innsbruck, Innsbruck, Austria
[8]Gold Brand Software, LLC, Herndon, VA, USA
[9]Department of Computer Science and Engineering, University of California, Santa Cruz, CA, USA
[10]Jacobs Engineering Group Inc., Herndon, VA, USA

MG, 0000-0002-8914-6887; MB, 0000-0002-6064-5316;
AD, 0000-0003-3989-9941; BG, 0000-0003-1915-9217;
FH, 0000-0001-9606-0427; MJ, 0000-0001-8759-6393;
CT, 0000-0003-2700-4557; TP, 0000-0002-0592-577X

The Defense Advanced Research Projects Agency (DARPA) programme 'Systematizing Confidence in Open Research and Evidence' (SCORE) aims to generate confidence scores for a large number of research claims from empirical studies in the social and behavioural sciences. The confidence scores will provide a quantitative assessment of how likely a claim will hold up in an independent replication. To create the scores, we follow earlier approaches and use prediction markets and surveys to forecast replication outcomes. Based on an initial set of forecasts for the overall replication rate in SCORE and its dependence on the academic discipline and the time of publication, we show that participants expect replication rates to increase over time. Moreover, they expect replication rates to differ between fields, with the highest replication rate in economics (average survey response 58%), and the lowest in psychology and in education

(average survey response of 42% for both fields). These results reveal insights into the academic community's views of the replication crisis, including for research fields for which no large-scale replication studies have been undertaken yet.

# 1. Introduction

Replication has long been established as a key practice in scientific research [1,2]. It plays a critical role in controlling the impact of sampling error, questionable research practices, publication bias and fraud [1,2]. An increase in the effort to replicate studies has been argued to help establishing credibility within a field [3]. Moreover, replications allow us to test if results generalize to a different or larger population and help to verify the underlying theory [2,4,5] and its scope [6]. Despite the importance of replications, there are practical and resource-related constraints that limit the extent to which replications are conducted [7].

Previous studies have shown that information about replication outcomes can be elicited from the research community [8–11]. This suggests that forecasting the outcomes of hypothetical replications can help assessing replication probabilities without requiring the resources for actually conducting replications. The Defense Advanced Research Projects Agency (DARPA) programme 'Systematizing Confidence in Open Research and Evidence' (SCORE) follows this approach to generate confidence scores for thousands of research claims from empirical studies in the social and behavioural sciences. These confidence scores will provide a quantitative assessment of how likely a claim will hold up in an independent replication. A small subset of the claims (about 5%) will eventually be assessed through replication, and the replication outcomes will be used to evaluate the accuracy of the confidence scores. The research claims are sampled from studies published during a 10 year period (2009–2018) across 60 journals from a number of academic disciplines.

To generate confidence scores for the DARPA SCORE programme [12], we follow the template of past forecasting studies [8–11] and use surveys and prediction markets. In the surveys, participants recruited from the relevant research communities are asked to provide their estimates for the probability that a claim will hold up in a replication. Replication here refers to either direct replication (i.e. same data collection process and analysis on a different sample) or data-analytic replication (i.e. same analysis on a similar but independent dataset; see [13,14]). Successful replication is defined as an effect that is in the same direction as the original effect and statistically significant at $p < 0.05$. While replication projects such as SCORE typically provide additional and non-binary characteristics of the replication results, such a binary definition is well-suited for elicitation of forecasts through prediction markets and surveys. In the prediction markets [15–17], participants trade contracts with pay-offs tied to the outcome of replications and thereby generate prices that provide quantitative forecasts of the replication results [8–11]. Because of the large number of claims to be assessed, forecasting takes place in monthly rounds from mid-2019 to mid-2020. In each monthly round, about 300 claims are assessed, resulting in about 3000 claims assessed by mid-2020.

Before we started collecting claim-specific forecasts, we collected an initial set of surveys and market forecasts for the overall replication rate in SCORE and its dependence on the academic discipline and the time of publication. These forecasts allow us to test whether participants at the beginning of the SCORE project expect field-specific and time-dependent variation of the replication rates. In this paper, we present an analysis of the data from this initial round of forecasting. Whether and to which extent these meta-forecasts are correct will be explored once the replications have actually been conducted.

# 2. Methods

The surveys and prediction markets to forecast time- and discipline-specific replication rates were open for one week each (12–18 August, 2019 and 19–25 August, 2019), with the market starting after the survey closed. We asked participants to forecast the overall SCORE replication rate, the replication rate in five non-overlapping 2-year periods (2009/2010, 2011/2012, 2013/2014, 2015/2016 and 2017/2018) and in six discipline clusters (economics, political science, psychology, education, sociology and criminology, and marketing, management and related areas). The discipline clusters are defined through journals (see the electronic supplementary material, table S1). Two of these clusters, namely sociology and criminology, and marketing, management and related areas, are heterogeneous in terms of research fields and comprise fields with a small number of journals sampled in SCORE. We combined those fields into clusters to meet a minimal number of journals per cluster. To elicit forecasts, we used the same wording in the survey and the prediction market (see the electronic supplementary material,

table S2). Further information, including the definition of what constitutes a successful replication, and the targeted power of the replications, were available in the online instructional material.

The surveys were incentivized using a peer assessment method which employs a surrogate scoring rule. Surrogate scoring rules [18] provide an unbiased estimate of strictly proper scoring rules [19] and are dominantly truthful in eliciting probabilistic predictions. When scoring an agent's predictions, surrogate scoring rules first construct a noisy prediction of the event's outcome from other agents' reports (noisy 'ground truth'). The second step estimates the bias of this noisy 'ground truth' using all elicited predictions across all users on all forecasting questions. Then, surrogate scoring rules compute a de-biased version of strictly proper scoring rules. Consequently, to maximize one's expected surrogate score, it is always a dominant strategy to report truthfully, owing to the incentive property of the strictly proper scoring rules. The total prize pool of $960 was allocated in prizes of $80 for the top six participants, $40 for participants ranked 7–12 and $20 for participants ranked 13–24.

In the prediction markets, the participants traded contracts that pay out points proportional to the actual replication rate in each time period and discipline cluster. The total prize pool of $4320 will be allocated once the replications are completed in proportion of the points earned from the contracts. To facilitate trading, we used a market maker implementing a logarithmic market scoring rule [16] with base 2 and a liquidity parameter of $b = 100$. Participants received an initial endowment of 100 points to trade with. Findings from previous replication-focused prediction markets [8–11] showed that most of the information elicited in these markets gets typically priced in soon after the markets opened; after about 2 days of trading, prices tend to fluctuate around equilibrium prices. To minimize the impact of noisy price fluctuations on our forecasts, we use a pre-registered time-weighted average of the prices from the second half of the trading period as market forecasts.

Participants were recruited through a number of mailing lists, Twitter and blog posts. As of the start of the initial surveys, 478 forecasters had signed up to participate and expressed informed consent, and 226 subsequently completed the initial forecasting survey. Most of these participants (80%) were from academia, with smaller groups from the private sector, non-profit organizations and the public sector (11%, 4% and 4%, respectively). The majority (69%) of participants are in early (e.g. undergraduate, graduate student, postdoc, assistant professor) or mid (e.g. senior research fellow) career stages. Those at senior career stages (e.g. full or emeritus professor) made up 7.5% of the survey takers. Self-reported interests of our participants were dominated by two fields: 99 survey respondents (44%) indicated that they were interested in economics and 126 (56%) in psychology. Marketing, management and related areas, the next largest field, was selected by 45 participants (20%), followed by political science, education, and sociology and criminology with 24 (11%), 23 (10%) and 22 (10%) responses, respectively. About half of all participants (122) only gave one field of interest, 65 (29%) respondents indicated two fields of interest, and 38 (17%) reported three or more fields of interest. Forty-eight per cent of the survey participants indicated that they had pre-registered at least one study before, and 34% have been involved in a replication study. About 37% of survey takers indicated that they had participated in a prediction market prior to SCORE. About 217 forecasters made at least one trade in the prediction markets. Given that 64% of these market traders also completed the survey, the participants in the market had a demographic composition similar to those who completed the survey.

The experimental design and statistical analyses were pre-registered on Open Science Framework (OSF), before the initial surveys and markets were opened. The data collected in the initial forecasting round allow us to analyse four of eight hypotheses provided in the pre-registration: (i) whether forecasted replication rates differ between fields of research; (ii) whether forecasted replication rates differ between time periods; (iii) whether topic-specific forecasts depend on the forecasters' field of research; and (iv) whether survey-based aggregated forecasts and market-based forecasts are correlated. For all analyses, we use survey responses; for analysis (iii), we additionally use responses to a demographic survey that included a question on academic interests; and for analysis (iv), we additionally use the prediction market data. All these analyses were conducted as pre-registered. The remaining four pre-registered analyses require data that become available once the forecasting for the individual research claims and the replications are completed. For statistical tests, we interpret the threshold of $p < 0.005$ as identifying statistical significance, and the threshold of $p < 0.05$ as identifying suggestive evidence [20]. Pre-registration document, data, codebook and scripts are available at the Dryad repository (https://dx.doi.org/10.5061/dryad.pg4f4qrk5) [21].

## 3. Results

Summary statistics for the surveys and markets are provided in table 1. The mean of the survey responses for the overall replication rate forecasts was 49%, which is close to the replication rate across previous

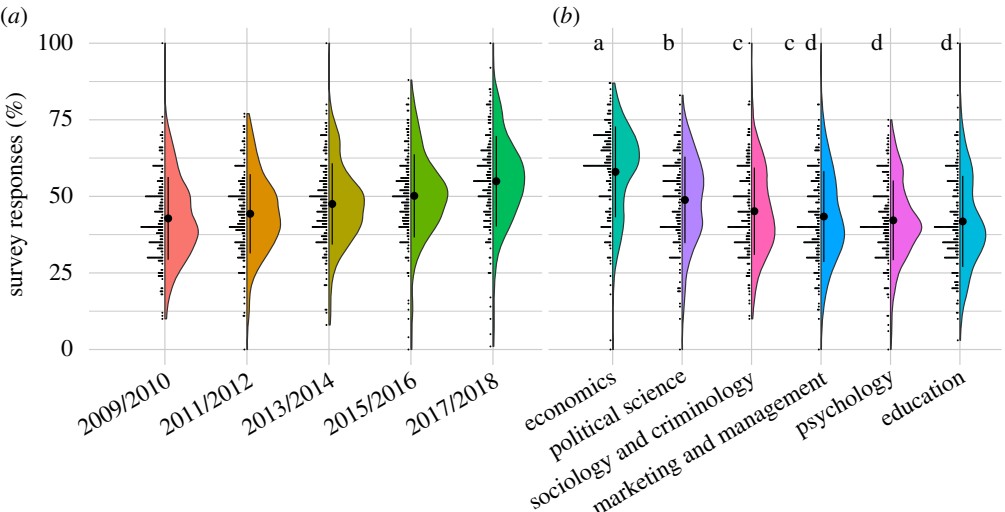

**Figure 1.** (*a*) Expected replication rate for publications from different 2 year periods. (*b*) Expected replication rate for publications from different fields. Points and error bars within the violin plots indicate the mean ± 1 s.d. Letters in (*b*) indicate significance grouping: fields with the same grouping label do not have significantly different means. Groupings are omitted for (*a*) as all time periods have statistically significant or suggestive differences.

**Table 1.** Descriptive statistics of survey and market forecasts. (Final price refers to the price at market closing, whereas smoothed price is a weighted average of the market prices, designed to reduce effects of noise near the end of the market. The aggregation methods are all highly correlated, with the Pearson's correlation coefficient between market and surrogate score rules (SSR) aggregation methods as follows: smoothed price and SSR brier 0.963, smoothed price and SSR rank 0.935, smoothed price and survey mean 0.957, SSR brier and SSR rank 0.924, SSR brier and survey mean 0.942, SSR rank and survey mean 0.979. All correlations are statistically significant at $p < 0.0001$ (d.f. = 10).)

| overall replication rate in… | smoothed price | final price | distinct traders | number of trades | survey mean | SSR rank | SSR brier |
|---|---|---|---|---|---|---|---|
| economics | 0.57 | 0.58 | 156 | 235 | 0.58 | 0.65 | 0.36 |
| political science | 0.45 | 0.46 | 115 | 158 | 0.49 | 0.55 | 0.29 |
| psychology | 0.41 | 0.45 | 165 | 226 | 0.42 | 0.39 | 0.26 |
| education | 0.39 | 0.45 | 126 | 176 | 0.42 | 0.38 | 0.24 |
| sociology and criminology | 0.44 | 0.43 | 106 | 133 | 0.45 | 0.45 | 0.29 |
| marketing and management | 0.40 | 0.36 | 124 | 161 | 0.43 | 0.41 | 0.25 |
| 2009/2010 | 0.41 | 0.40 | 74 | 107 | 0.43 | 0.41 | 0.25 |
| 2011/2012 | 0.43 | 0.42 | 60 | 77 | 0.44 | 0.44 | 0.26 |
| 2013/2014 | 0.45 | 0.44 | 64 | 80 | 0.48 | 0.49 | 0.27 |
| 2015/2016 | 0.45 | 0.46 | 98 | 147 | 0.50 | 0.53 | 0.30 |
| 2017/2018 | 0.50 | 0.49 | 82 | 104 | 0.55 | 0.58 | 0.32 |
| all claims in SCORE | 0.47 | 0.48 | 83 | 125 | 0.49 | 0.49 | 0.31 |

replication projects [8,9,22,23]. The survey results provide evidence that the participants expect the replications rates to differ across the 2 year time periods (one-way repeated measures ANOVA; $F_{4,900} = 130.5$, $p < 0.0001$). Replication rates were expected to increase over time from 43% in 2009/2010 to 55% in 2017/2018 (figure 1*a*). Using pairwise paired *t*-tests, with a Benjamini–Hochberg correction to control for the false discovery rate, all year bands were forecasted to have different replication rates significant at the 0.0001 level, except for the comparison between 2009/2010 and 2011/2012, where suggestive evidence is found for differences in replication rate expectations ($p = 0.0066$; see the electronic the supplementary material, table S3 for all *p*-values and test statistics).

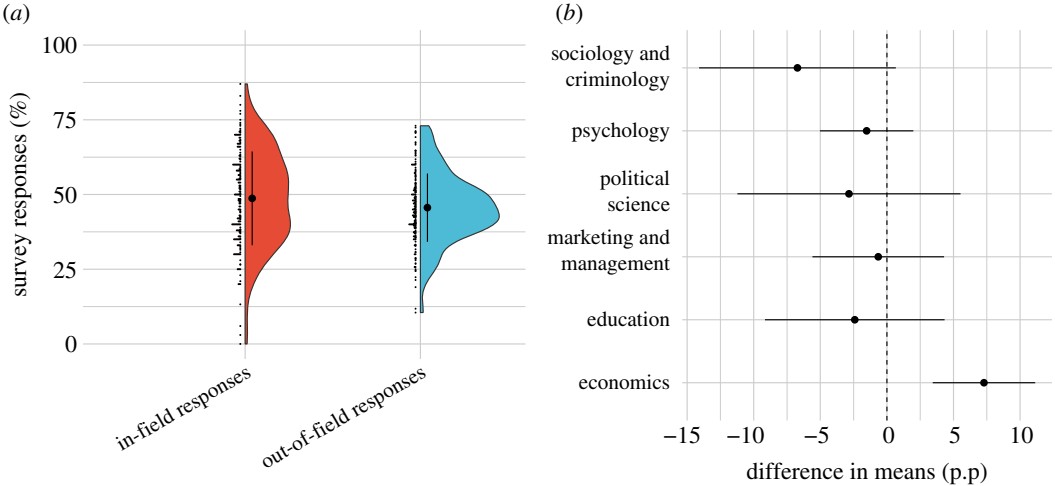

**Figure 2.** (a) In-field versus out-of-field responses. Participants predict a higher replication rate for their fields of interest, as compared to other fields. (b) Difference of evaluation of a field by in-field and out-field participants (in per cent points). Participants with interest in economics predict a higher replication rate for this field compared to participants with no interest in economics. For other fields, such an effect is not observed. Points and error bars indicate the mean ± 1 s.d.

The survey responses also show that participants expected replication rates to differ between topics. Using responses from the survey, a one-way repeated measures ANOVA finds a joint significant effect of topic variables ($F_{5,1225} = 82.02$, $p < 0.0001$). Using pairwise paired $t$-tests, and the Benjamini–Hochberg correction to control for the false discovery rate, 9 of the 15 topic-topic pairs are found to have a statistically significant difference ($p$-value $< 0.0001$) in mean expected replication rates. Results including significance groupings are reported in figure 1b. The participants expect the highest replication rate in economics (average response 58%) and the lowest in psychology and in education (average response of 42% for both fields). The complete results are given in the electronic supplementary material, table S4.

We observe that the forecasters' average responses for topics which match their own fields of interest are higher than their average responses for topics not belonging to their fields of interest (paired $t$-test, $t_{197} = -3.35$, $p = 0.0010$). The average response for 'in-field' topics is 48.7%, compared to 45.6% for 'out-of-field' topics (figure 2a). To identify the mechanisms behind this finding, we followed up with two additional analyses that were not pre-registered. To test if research fields are more optimistically assessed by participants with interest in this field, as compared to participants with no interest in this field, we performed unpaired $t$-tests, comparing 'in-field' responses with 'out-of-field' responses for each topic separately. The only statistically significant effect is found in economics. Participants interested in economics were more optimistic about the replication rates in economics than those not interested in economics (unpaired $t$-test, $t_{199.85} = 3.74$, $p = 0.0002$). No evidence was found for such an effect within the other topics (figure 2b). Moreover, we investigated how the forecast for the overall replication rate depends on demographic characteristics. Suggestive evidence is observed for only one of the demographic variables included: participants who stated that they have been involved in a replication study before on average forecasted a lower overall replication rate ($t_{217} = -2.75$, $p = 0.006$). Effect sizes and test statistics are given in the electronic supplementary material, table S5.

A key part of our experimental design for SCORE is the use of two alternative methodologies for eliciting and aggregating forecasts. Although there is an overlap of participants involved in survey and markets, the methodologies are independent and differ from each other in terms of elicitation and aggregation of information. To test if both methodologies yield similar results, we performed a correlation test between our two main aggregators: the smoothed market price and the forecasts generated by the survey-based peer assessment method. The Pearson correlation coefficient is 0.935 (d.f. = 10, $p < 0.0001$), supporting that our findings are robust with respect to the elicitation and aggregation methodology.

## 4. Discussion

Forecasting research outcomes has been argued to benefit science [15,24,25]. Hanson [15] suggested using prediction markets to forecast research outcomes to more efficiently reach a consensus on scientific

questions and counteract inaccurate but popular beliefs. He also pointed out that funders could use prediction markets to incentivize research on questions they prioritize, without having to commit funding to specific research groups. Moreover, ex-ante predictions from prediction markets could help set priors for Bayesian statistical inference, prioritize research questions for hypotheses testing [24], and help to better capture how novel or surprising a result is [25].

The aim of our previous forecasting projects [8–11] was to test whether within the research communities there is information about the replicability of studies, and whether surveys and prediction markets can aggregate this information into accurate forecasts. The results of these previous studies were encouraging: the forecasted probabilities were informative with respect to the observed replication outcomes. For the SCORE project, we go beyond such a proof-of-principle. We elicit information on a large set of research claims with only a small subset being evaluated through replication of reproduction. This approach illustrates how the information gained from the forecasting can be scaled up without necessarily scaling up cost-intensive replications.

The forecasts presented in this study focus on field-specific and time-specific replication rates, rather than the probability of replication for individual claims. Previous forecasting studies have shown that while forecasts for single claims are informative with respect to observed replication outcomes, they tend to be too optimistic. Explicit forecasts for overall replication rates might be more reliable than what can be inferred from forecasts on individual replications; this is a hypothesis we have pre-registered to test once the results from the SCORE replications are available.

Our results show that participants expect replication rates to increase over time, from 43% in 2009/2010 to 55% in 2017/2018. The reasons behind this expectation have not been elicited in our study and are an interesting topic for future research. One plausible explanation might be that participants expect recent methodological changes in the social and behavioural sciences to have a positive impact on replication rates. This is also in line with an increased creation and use of study registries during this time period in the social and behavioural science [3]. Further insights into longitudinal patterns in the reliability of published research could follow from replication projects on studies sampled across an extended period of time.

Similarly, the observed differences in topic-specific expected replication rates deserve further study. For fields that have been covered by replication studies in the past, the expected replication rates are probably anchored around past replication projects; the point estimate of the replication rate in the Replication Project: psychology [23], for instance was lower than in the Experimental Economics Replication Project [8], although it should be noted that the inclusion criteria, time periods and sample sizes differed between these projects and thus it is not straightforward to compare these numbers. Differences in expected replication rates could further reflect that hypotheses and typical effect sizes differ between fields, different fields employ different methodologies and policies, and results from different fields might be subjected to different biases.

Because forecasts from previous replication reports were informative with respect to replication outcomes, the forecasts presented here might provide some guidance on how credible claims in different subjects are. This is particularly the case for fields for which no other information is available, i.e. fields with no past large-scale replication project such as education, political science and marketing, management and related areas. If our forecasts hold up, it will be interesting to investigate if specific factors (such as different methodologies and policies) can be identified that influence replication rates.

Ethics. We gratefully acknowledge review and approval of the study design through DOD HRPO and Harvard University CUHS.

Data accessibility. All data and code have been submitted to Dryad https://doi.org/10.5061/dryad.pg4f4qrk5 [21].

The pre-analysis plan has been submitted to OSF (osf.io/w4xsk/), where it is under embargo until completion of the SCORE project. A copy of the relevant pre-registration document has been released on the Dryad repository.

Authors' contributions. All authors contributed to designing the experiment and drafting the manuscript. D.V., M.G. and T.P. developed the pre-analysis plan; C.T., Y.C. and T.P. coordinated the experiments; M.B., B.G. and C.T. conducted the experiment; M.G., D.V., M.B., Y.C., Y.L. and J.W. contributed to the data analysis; M.G. and D.V. carried out the statistical analyses.

Competing interests. We declare we have no competing interests.

Funding. This material is based upon work supported by the Defense Advanced Research Projects Agency (DARPA) and Space and Naval Warfare Systems Center Pacific (SSC Pacific) under contract no. N66001-19-C-4014. T.P. thanks the Marsden Fund for financial support for project MAU-1710.

Acknowledgements. We gratefully acknowledge review and approval of the study design through DOD HRPO and Harvard University CUHS.

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
