## [Reviewer comments · Royal Society Open Science]

Review History

RSOS-200566.R0 (Original submission)

Review form: Reviewer 1 (Michèle Nuijten)

Is the manuscript scientifically sound in its present form?

Yes

Are the interpretations and conclusions justified by the results?

Yes

Is the language acceptable?

Yes

Do you have any ethical concerns with this paper?

No

Have you any concerns about statistical analyses in this paper?

No

Recommendation?

Accept with minor revision (please list in comments)

Comments to the Author(s)

This manuscript presents the results of a survey and prediction market to assess expected replicability in different scientific fields and time periods. I think the manuscript is interesting and well-written. I only have a few questions/suggestions for the authors to consider.

First, I may have overlooked it, but I missed a definition of a successful replication. This is crucial information, as it affects the interpretation of the expected replication rates. Does a replication rate of .46 mean that the participants expected that (on average) 46% of the replicated studies will show a statistically significant effect in the same direction as the original? Or did the authors maintain another definition? Please introduce the definition in the manuscript, and report whether this definition was also given to the participants or not. If not, the results will be more difficult to interpret, as there are many different opinions on what constitutes a successful replication.

Second, I applaud the authors for preregistering their study. Unfortunately, however, I did not have access to the OSF page, so I could not evaluate the preregistration. I'm assuming that the preregistration will be made public in time (after all, the fact that the study was preregistered has little meaning if readers do not have access to the preregistration). Beside openly publishing the preregistration, I would also like to ask the authors to explicitly state in the manuscript whether all the reported analyses (except the one additional exploratory analysis they already identified as such) were performed (exactly) according to the preregistration, and whether all preregistered analyses have been conducted.

Third, it is great that the authors shared their data. I would like to ask them to also include a codebook that describes the variables in the datasets they uploaded. The reproducibility of the manuscript would also significantly increase if the authors would also upload their analysis scripts.

Please find some final, minor remarks below.

Signed,
Michèle Nuijten

Minor remarks

- * P. 2, line 45: I would change the wording to better reflect the chronological order of the different steps: "i.e., data collection process and same analysis on a different sample"
- * Please also report test statistics and degrees of freedom in the text, in addition to the p-values
- * p. 3: from the text, I can't really figure out what this Surrogate Scoring Rule is. Would it be possible to add a couple of sentences to explain this in a bit more detail?
- * Typo in caption Figure 2: "Participants with interest is economics"

Review form: Reviewer 2**Is the manuscript scientifically sound in its present form?**

No

Are the interpretations and conclusions justified by the results?

No

Is the language acceptable?

Yes

Do you have any ethical concerns with this paper?

No

Have you any concerns about statistical analyses in this paper?

No

Recommendation?

Reject

Comments to the Author(s)

Using surveys and prediction markets to estimate replicability in science is an open question but a worthwhile research endeavor. However, the question and results of field-level replicability estimates provides little to no useful information. I do not see the “unique insight” that the authors claim.

The primary question investigated resides in the title: are replication rates the same across academic fields? The authors use prediction markets and surveys to forecast whether replication outcomes across various fields in the social sciences. Unsurprisingly, participants expect that replication rates differ between fields and that replication rates increase over time.

What is learned from this question and the results? Would it have been unique to find that participants thought fields had the same replication rates? What do we learn either way? We don't have replication rates for many of the fields, as noted by the authors. So what do we learn today from this finding? It is like asking participants to predict which field of medicine will have the most number of Nobel prizes by 2030. I guess the survey results are interesting in the same way that a USA Today survey about people's guesses as to who will win the NBA finals next year is interesting. Unless the 12 authors on this paper can do a better job explaining what is learned, it remains a mystery what the unique insights are.

The most unsupported claim in this paper is the final paragraph. It states that this study “serves as [sic] a informative guide to the credibility of claims from these fields.” The study does not support this claim in any way. Policy makers and readers should NOT be using this study and its results to assess replication rates in Education, Political Science, etc. This conclusion does not fall from the results and analysis of this paper.

In the discussion, the authors stated that the “forecasted probabilities were informative with respect to the observed replication outcomes.” But weren't many of the fields' replication rates unknown? I am assuming that this is for the fields that we do have replication rates for. If so, wouldn't researchers willing to participate in a study about replication rates be somewhat aware already of the replication rates in these various fields. Saying that participants overall came pretty close to the replication rates of the field is like asking public health officials to estimate COVID infections across states in the U.S. and then finding that responses are pretty close to the actual numbers. It doesn't mean that the crowd mysteriously can predict replication rates in psychology and economics.

One of the few semi-interesting results was that forecasters seemed to think a little bit higher of their own field but not by much. The in-field topics was 48.7% where the out-of-field was 45.6%. Does this difference mean anything practically? Which fields were affected by this the most? Why were they affected the most? There is very little information about the participants themselves. This is critical to understanding these results. The reader is told that most of the participants were early career scientists. What else is known about these participants and how do these known features interact with the in-field results? Again, I am lost on what is learned from the questions asked in this paper and supposed answers to those questions.

Decision letter (RSOS-200566.R0)

Dear Dr Pfeiffer,

The editors assigned to your paper ("Are replication rates the same across academic fields? Community forecasts from the DARPA SCORE program") have now received comments from reviewers. We would like you to revise your paper in accordance with the referee and Associate Editor suggestions which can be found below (not including confidential reports to the Editor). Please note this decision does not guarantee eventual acceptance.

Please submit a copy of your revised paper before 21-Jun-2020. Please note that the revision deadline will expire at 00.00am on this date. If we do not hear from you within this time then it will be assumed that the paper has been withdrawn. In exceptional circumstances, extensions may be possible if agreed with the Editorial Office in advance. We do not allow multiple rounds of revision so we urge you to make every effort to fully address all of the comments at this stage. If deemed necessary by the Editors, your manuscript will be sent back to one or more of the original reviewers for assessment. If the original reviewers are not available, we may invite new reviewers.

- Data accessibility

If you wish to submit your supporting data or code to Dryad (<http://datadryad.org/>), or modify your current submission to dryad, please use the following link:
<http://datadryad.org/submit?journalID=RSOS&manu=RSOS-200566>

- Competing interests

- Authors' contributions

- Acknowledgements

- Funding statement

on behalf of Professor Zoltan Dienes (Associate Editor) and Mark Chaplain (Subject Editor)
openscience@royalsociety.org

Reviewers' Comments to Author:

Reviewer: 1

Comments to the Author(s)

This manuscript presents the results of a survey and prediction market to assess expected replicability in different scientific fields and time periods. I think the manuscript is interesting and well-written. I only have a few questions/suggestions for the authors to consider.

First, I may have overlooked it, but I missed a definition of a successful replication. This is crucial information, as it affects the interpretation of the expected replication rates. Does a replication rate of .46 mean that the participants expected that (on average) 46% of the replicated studies will show a statistically significant effect in the same direction as the original? Or did the authors maintain another definition? Please introduce the definition in the manuscript, and report whether this definition was also given to the participants or not. If not, the results will be more difficult to interpret, as there are many different opinions on what constitutes a successful replication.

Second, I applaud the authors for preregistering their study. Unfortunately, however, I did not have access to the OSF page, so I could not evaluate the preregistration. I'm assuming that the preregistration will be made public in time (after all, the fact that the study was preregistered has little meaning if readers do not have access to the preregistration). Beside openly publishing the preregistration, I would also like to ask the authors to explicitly state in the manuscript whether all the reported analyses (except the one additional exploratory analysis they already identified as such) were performed (exactly) according to the preregistration, and whether all preregistered analyses have been conducted.

Third, it is great that the authors shared their data. I would like to ask them to also include a codebook that describes the variables in the datasets they uploaded. The reproducibility of the manuscript would also significantly increase if the authors would also upload their analysis scripts.

Please find some final, minor remarks below.

Signed,
Michèle Nuijten

Minor remarks

- * P. 2, line 45: I would change the wording to better reflect the chronological order of the different steps: "i.e., data collection process and same analysis on a different sample"
- * Please also report test statistics and degrees of freedom in the text, in addition to the p-values
- * p. 3: from the text, I can't really figure out what this Surrogate Scoring Rule is. Would it be possible to add a couple of sentences to explain this in a bit more detail?
- * Typo in caption Figure 2: "Participants with interest is economics"

Reviewer: 2

Comments to the Author(s)

Using surveys and prediction markets to estimate replicability in science is an open question but a worthwhile research endeavor. However, the question and results of field-level replicability estimates provides little to no useful information. I do not see the "unique insight" that the authors claim.

The primary question investigated resides in the title: are replication rates the same across academic fields? The authors use prediction markets and surveys to forecast whether replication outcomes across various fields in the social sciences. Unsurprisingly, participants expect that replication rates differ between fields and that replication rates increase over time.

What is learned from this question and the results? Would it have been unique to find that participants thought fields had the same replication rates? What do we learn either way? We don't have replication rates for many of the fields, as noted by the authors. So what do we learn today from this finding? It is like asking participants to predict which field of medicine will have the most number of Nobel prizes by 2030. I guess the survey results are interesting in the same way that a USA Today survey about people's guesses as to who will win the NBA finals next year

is interesting. Unless the 12 authors on this paper can do a better job explaining what is learned, it remains a mystery what the unique insights are.

The most unsupported claim in this paper is the final paragraph. It states that this study “serves as [sic] a informative guide to the credibility of claims from these fields.” The study does not support this claim in any way. Policy makers and readers should NOT be using this study and its results to assess replication rates in Education, Political Science, etc. This conclusion does not fall from the results and analysis of this paper.

In the discussion, the authors stated that the “forecasted probabilities were informative with respect to the observed replication outcomes.” But weren’t many of the fields’ replication rates unknown? I am assuming that this is for the fields that we do have replication rates for. If so, wouldn’t researchers willing to participate in a study about replication rates be somewhat aware already of the replication rates in these various fields. Saying that participants overall came pretty close to the replication rates of the field is like asking public health officials to estimate COVID infections across states in the U.S. and then finding that responses are pretty close to the actual numbers. It doesn’t mean that the crowd mysteriously can predict replication rates in psychology and economics.

One of the few semi-interesting results was that forecasters seemed to think a little bit higher of their own field but not by much. The in-field topics was 48.7% where the out-of-field was 45.6%. Does this difference mean anything practically? Which fields were affected by this the most? Why were they affected the most? There is very little information about the participants themselves. This is critical to understanding these results. The reader is told that most of the participants were early career scientists. What else is known about these participants and how do these known features interact with the in-field results? Again, I am lost on what is learned from the questions asked in this paper and supposed answers to those questions.

Author's Response to Decision Letter for (RSOS-200566.R0)

See Appendix A.

Decision letter (RSOS-200566.R1)

Dear Dr Pfeiffer,

It is a pleasure to accept your manuscript entitled "Are replication rates the same across academic fields? Community forecasts from the DARPA SCORE program" in its current form for publication in Royal Society Open Science. The comments of the reviewer(s) who reviewed your manuscript are included at the foot of this letter.

on behalf of Professor Zoltan Dienes (Associate Editor) and Mark Chaplain (Subject Editor)
openscience@royalsociety.org

Appendix A

Reviewer: 1

This manuscript presents the results of a survey and prediction market to assess expected replicability in different scientific fields and time periods. I think the manuscript is interesting and well-written. I only have a few questions/suggestions for the authors to consider.

First, I may have overlooked it, but I missed a definition of a successful replication. This is crucial information, as it affects the interpretation of the expected replication rates. Does a replication rate of .46 mean that the participants expected that (on average) 46% of the replicated studies will show a statistically significant effect in the same direction as the original? Or did the authors maintain another definition? Please introduce the definition in the manuscript, and report whether this definition was also given to the participants or not. If not, the results will be more difficult to interpret, as there are many different opinions on what constitutes a successful replication.

We thank the reviewer for their comments. We define successful replication as statistically significant finding in the direction of the original claim, and thus interpret a replication rate of .46 as expectation that 46% of the replicated studies will show a statistically significant effect in the same direction. The definition of a successful replication is now included in the manuscript, and was made available to the participants in the online instructional material, together with targets for the power of the replications.

Second, I applaud the authors for preregistering their study. Unfortunately, however, I did not have access to the OSF page, so I could not evaluate the preregistration. I'm assuming that the preregistration will be made public in time (after all, the fact that the study was preregistered has little meaning if readers do not have access to the preregistration). Beside openly publishing the preregistration, I would also like to ask the authors to explicitly state in the manuscript whether all the reported analyses (except the one additional exploratory analysis they already identified as such) were performed (exactly) according to the preregistration, and whether all preregistered analyses have been conducted.

The pre-registration with OSF is currently embargoed, with the embargo being lifted once the entire SCORE project is completed. Unfortunately, OSF does not support to lift the embargo on a single document. To make the pre-registration document for the initial forecasting round publicly available, we therefore decided to release it on the dryad repository, together with the data and the scripts. Of the eight hypotheses that were pre-registered we can at this point test four. The tests were conducted exactly as pre-registered. The remaining tests require data available once the SCORE project is completed. This is clarified in the manuscript.

Third, it is great that the authors shared their data. I would like to ask them to also include a codebook that describes the variables in the datasets they uploaded. The reproducibility of the manuscript would also significantly increase if the authors would also upload their analysis scripts.

We made a codebook and the script available on the dryad repository.

Minor remarks

* P. 2, line 45: I would change the wording to better reflect the chronological order of the different steps: “i.e., data collection process and same analysis on a different sample”

This has been corrected.

* Please also report test statistics and degrees of freedom in the text, in addition to the p-values

We now give test statistics and degrees of freedom in the text.

* p. 3: from the text, I can't really figure out what this Surrogate Scoring Rule is. Would it be possible to add a couple of sentences to explain this in a bit more detail?

We added more detailed explanations on surrogate scoring.

* Typo in caption Figure 2: “Participants with interest is economics”

This has been corrected.

Reviewer: 2

Using surveys and prediction markets to estimate replicability in science is an open question but a worthwhile research endeavor. However, the question and results of field-level replicability estimates provides little to no useful information. I do not see the “unique insight” that the authors claim.

The primary question investigated resides in the title: are replication rates the same across academic fields? The authors use prediction markets and surveys to forecast whether replication outcomes across various fields in the social sciences. Unsurprisingly, participants expect that replication rates differ between fields and that replication rates increase over time.

What is learned from this question and the results? Would it have been unique to find that participants thought fields had the same replication rates? What do we learn either way? We don't have replication rates for many of the fields, as noted by the authors. So what do we learn today from this finding? It is like asking participants to predict which field of medicine will have the most number of Nobel prizes by 2030. I guess the survey results are interesting in the same way that a USA Today survey about people's guesses as to who will win the NBA finals next year is interesting. Unless the 12 authors on this paper can do a better job explaining what is learned, it remains a mystery what the unique insights are.

We thank the reviewer for their comments. We revised the introduction and discussion to detail why these findings are of interest. While we believe that our findings are unique in that we are not aware of any other study that elicits estimates of replication rates across fields, we drop the term ‘unique’ from the abstract as this is an unnecessary intensifier.

The most unsupported claim in this paper is the final paragraph. It states that this study “serves as [sic] a informative guide to the credibility of claims from these fields.” The study does not support this claim in any way. Policy makers and readers should NOT be using this study and its results to assess replication rates in Education, Political Science, etc. This conclusion does not fall from the results and analysis of this paper.

We rephrased the final paragraph to provide a more fine-grained description how our findings could help inform policies. Documenting that expectations for replications rates differ between fields raises a number of follow-up questions on why such an expectation exists, and whether this expectation is correct. Further research is required to answer these questions and can help inform policies to improve the reliability of published research.

In the discussion, the authors stated that the “forecasted probabilities were informative with respect to the observed replication outcomes.” But weren’t many of the fields’ replication rates unknown? I am assuming that this is for the fields that we do have replication rates for. If so, wouldn’t researchers willing to participate in a study about replication rates be somewhat aware already of the replication rates in these various fields. Saying that participants overall came pretty close to the replication rates of the field is like asking public health officials to estimate COVID infections across states in the U.S. and then finding that responses are pretty close to the actual numbers. It doesn’t mean that the crowd mysteriously can predict replication rates in psychology and economics.

We rephrased and expanded this section in the discussion to clarify the points we are making in this context. The statement that “forecasted probabilities were informative with respect to the observed replication outcomes” refers our past forecasting projects. In the SCORE project, we elicit forecasts on 3000 individual research claims from across the behavioural and social sciences. This is work in progress, and the individual forecasting being completed around 08/2020 and the final replication results being available at the end of 2020. The data presented in the submitted manuscript are from forecasts for the overall replication rates.

One of the few semi-interesting results was that forecasters seemed to think a little bit higher of their own field but not by much. The in-field topics was 48.7% where the out-of-field was 45.6%. Does this difference mean anything practically? Which fields were affected by this the most? Why were they affected the most? There is very little information about the participants themselves. This is critical to understanding these results. The reader is told that most of the participants were early career scientists. What else is known about these participants and how do these known features interact with the in-field results? Again, I am lost on what is learned from the questions asked in this paper and supposed answers to those questions.

The participant cohort is described in the methods section. In Fig 2 B we show that this is mainly driven by economists being more optimistic about economics. Since economists are the largest cohort in the study, economics is affected most. We added an additional analysis to investigate the relation between demographics and overall forecast for the replication rate in SCORE. We find suggestive evidence for participants who previously participated in replication studies to provide lower estimates for the overall replication rate. This finding is added to the results section; the full analysis is provided in SI Table 6.